# Inferring the Invisible: Recurrent Neuro-Symbolic Forward Chaining Network

## Abstract

A key challenge in artificial intelligence is inferring underlying factors that are not directly observable but are crucial for understanding and predicting complex behaviors. In this paper, we introduce a novel neural-symbolic framework that advances beyond traditional rule induction by integrating latent predicate discovery with rule learning. Our approach utilizes a recurrent unit to iteratively refine and learn rules from observed data, employing dynamic programming techniques to identify fixed points and solve complex problems. This framework enables the discovery of hidden predicates—such as user engagement or underlying motivations—that influence observable outcomes but are not directly grounded in the data. By encoding both explicit and latent predicates into a unified rule embedding, our method facilitates a deeper understanding of complex phenomena and enhances predictive accuracy. This joint learning process captures explicit relationships and invents new predicates essential for comprehensive inference. We validate our method across various tasks, demonstrating its capability to reveal hidden structures and enhance symbolic reasoning with deeper, more accurate insights.

## 1 Introduction

Neural-symbolic reasoning integrates neural networks' ability to learn complex patterns from data with symbolic reasoning's precision in applying logical rules Hitzler & Sarker (2022). This hybrid approach creates AI systems capable of not only detecting intricate patterns in unstructured data but also reasoning about them in a structured, explainable manner Yang et al. (2024).

Traditional rule induction methods focus on extracting explicit patterns from data but often fail to uncover latent predicates—hidden variables or relationships that are not directly observable Campero et al. (2018)Claire Glanois (2022). These methods are effective for learning surface-level rules but struggle with identifying underlying factors that drive complex phenomena. For example, in healthcare diagnostics, critical latent conditions must be inferred from incomplete or noisy data, and predicate invention is crucial for discovering these hidden factors and generating new, abstract concepts. Probabilistic models like Markov Logic Networks (MLNs) Richardson & Domingos (2006) can infer latent predicates but typically assume that the logical rules are predefined, which can limit adaptability and lead to computational challenges Oltramari et al. (2020).

Our framework tackles this gap by jointly learn the logical rules and infer (and discover) the hidden predicates. We employ a specialized recurrent unit, which we call the *Recurrent Neuro-Symbolic Forward Chaining Network (RNS-FCN)*. Our RNS-FCN operates in two distinct phases: forward inference, where the current logical rules are applied recursively to make the inference, and backpropagation, where the model refines and updates its logical rules based on error signals. The forward inference is achieved by running the recurrent unit through several steps until the states converge to a fixed point, making latent predicate inference straightforward. Rule learning is achieved in a differentiable manner, allowing for adaptive updates based on error signals and ensuring both transparency and flexibility in the model.

Our RNS-FCN shares some similarities with traditional Recurrent Neural Networks (RNNs) Grossberg (2013). Like RNNs, we utilize recurrent units where each layer reuses the same set of model parameters across steps. However, the depth of our model refers to more than just the number of

neural layers—it represents *deeper reasoning* akin to human thought processes. Much like how humans engage in iterative reasoning, refining their understanding as new evidence emerges. Think of it as a process of deeper thinking: as our model progresses through multiple layers, it is reasoning more deeply about the logical connections in the data, adjusting its beliefs until it reaches a stable conclusion Kuang et al. (2024). This contrasts with traditional deep learning, where increased depth typically refers to larger networks, but without a logical reasoning process guiding the layers. In this way, our model simulates recursive reasoning, pushing beyond surface-level pattern recognition and engaging in multi-step inferencing akin to human thought, offering a novel depth of interpretability and precision.

Each recurrent layer in the RNS-FCN performs a one-step inference guided by the current set of logical rules. The rules themselves, represented by the parameters of the recurrent units, are updated not during the inference phase but through backpropagation. In the forward computation, these rules act as constraints, shaping how the model updates its beliefs or hypotheses about the data at each step. The iterative process continues until the model reaches a fixed point where its inferences stabilize and no further updates are required. In the backward computation, backpropagation via gradient descent adjusts the model's parameters based on the discrepancy between predicted and actual outcomes, ensuring that the learned logical rules improve with each iteration. The backpropagation mechanism allows us to optimize the rule set in an end-to-end differentiable way.

In summary, our framework bridges the gap between traditional rule induction and the modeling of latent variables. The RNS-FCN framework provides a robust tool for discovering hidden structures and generating new concepts, offering a comprehensive solution for tasks that require deep understanding and adaptability. This end-to-end approach enhances our ability to uncover latent factors, making it invaluable for applications ranging from user behavior analysis to advanced decision-making systems in AI.

## 2 RELATED WORK

**Traditional ILP Methods**   Inductive Logic Programming (ILP) focuses on learning logical rules from relational data. Some papers rely on heuristic methods for efficiency. Cohen (1995) proposed *RIPPER*, a fast rule induction algorithm that builds rules iteratively using a separate-and-conquer strategy. Similarly, Quinlan (1990) developed *FOIL*, which operates by iteratively generating clauses that define target relations based on a training set of examples. It uses a greedy algorithm to add literals (conditions) to the rules, aiming to generalize the target relation effectively. The method presented in Dash et al. (2018) learns Boolean decision rules in disjunctive normal form (DNF) or conjunctive normal form (CNF) for binary classification. The method uses *column generation (CG)* to efficiently search through the exponentially large space of possible clauses. Wei et al. (2019) propose Generalized Linear Rule Models (GLRM), which integrate decision rules into generalized linear models (GLM) to balance interpretability and accuracy. The method formulates the rule learning process as an optimization problem, trading off rule complexity and predictive performance. Cropper & Morel (2023) propose an ILP approach called Learning from Failures (LFF), which consists of three stages: generate, test, and constrain. In the generate stage, a hypothesis is created based on syntactic constraints. In the test stage, the hypothesis is evaluated against training examples. The method is implemented in Popper, which integrates Answer Set Programming (ASP) and Prolog to learn optimal and recursive logic programs efficiently. These approaches demonstrate the reliance of ILP and related methods on heuristics to manage complexity. However, they generally may not guarantee globally optimal solutions. Pellegrina & Vandin (2024) propose SamRuLe, a scalable algorithm for learning nearly optimal rule lists via sampling. This method leverages VC-dimension bounds to ensure that the learned rule lists are close to the optimal solution on the full dataset. However, due to its reliance on sampling, SamRuLe may not handle noisy data effectively.

**ILP Methods with Differentiable Model**   Traditional Inductive Logic Programming (ILP) models struggle with noisy data and scalability. Differentiable approaches address these issues by integrating continuous relaxation, which allows gradient descent for optimization. Evans & Grefenstette (2018) proposed $\partial ILP$, which represents logic rules in a differentiable form and combines neural networks with symbolic logic. Manhaeve et al. (2018) introduced *DeepProbLog*, which extends ProbLog by integrating neural predicates for reasoning over both symbolic and subsymbolic data. While these differentiable ILP methods improve robustness to noise and enable joint optimization,

they face challenges in scalability and computational cost. Neural Logic Machines (NLMs) (Dong et al., 2019) combine MLPs with logic programming to perform inductive learning and logical reasoning. However, the implicit representation of rules in the network weights reduces interpretability.

**ILP Methods with Neural Embedding**  Embedding-based models are widely used for Knowledge Base (KB) completion. These models typically represent entities as low-dimensional vectors, while relations are modeled as linear or bilinear operators applied to these entities. Early models like TransE (Bordes et al., 2013) interpret relations as translations in the vector space but struggle with complex relations. TransH (Wang et al., 2014) and TransR (Lin et al., 2015) address this by projecting entities onto hyperplanes or distinct relation-specific spaces, improving the handling of diverse relations. More expressive models, such as Neural Tensor Networks (NTN) (Socher et al., 2013) and RESCAL (Nickel et al., 2011), capture higher-order interactions at the cost of increased complexity. ComplEx (Trouillon et al., 2016) introduces complex-valued embeddings for asymmetric relations, while multi-hop reasoning methods like Guu et al. (2015) (Guu et al., 2015) leverage path-based embeddings for traversing knowledge graphs. However, these approaches often face limitations in reasoning power, particularly in multi-hop scenarios.

Recent advances in inductive logic programming (ILP) are significantly influenced by Rocktäschel & Riedel (2017), who propose *Neural Theorem Proving (NTP)*. NTP integrates symbolic logic with neural networks through differentiable backward-chaining reasoning. Building on this, Campero et al. (2018) introduces a neural forward-chaining differentiable rule induction network. However, both approaches still rely on carefully hand-designed templates for each ILP task, which limits scalability. Claire Glanois (2022) advance these models by incorporating a hierarchical structure with an expressive set of meta-rules, enabling more flexible rule induction without the need for manual template design. Nevertheless, the model struggles to discover latent variables or invent new predicates, limiting its ability to uncover hidden patterns in complex reasoning tasks.

## 3  BACKGROUND

**Predicate**  In the context of logic-based AI systems, a predicate is a fundamental Boolean logic variable used to describe properties of or relationships between entities. Predicate variables are grounded by data, being True or False, and are served as the basic building blocks for logical expressions. For instance, a predicate like $Has\_Fever(Patient)$ denotes whether a patient has a fever, while $Use\_Drug(Patient)$ specifies whether a drug treats a particular patient. These predicates capture essential aspects of the system's state and relationships.

**Logic rules**  are formal constructs used to infer new knowledge based on the relationships among predicates. A prevalent form of logic rule is the **Horn clause**, which is expressed as:

$$f: \quad Q \leftarrow P_1 \wedge P_2 \wedge \cdots \wedge P_h \tag{1}$$

where $P_1, P_2, \ldots, P_h$ are predicates constituting the *body*, representing conditions that must be satisfied, while $Q$ is a predicate in the *head*, representing the conclusion. The rule indicates that if all predicates in the body hold true, the predicate $Q$ can be inferred. Horn rules are integral to logical reasoning systems, enabling the derivation of new facts from existing knowledge.

**Latent Predicates and Rule Learning**  In many complex domains, such as healthcare, finance, or social networks, not all predicates are known a priori. Latent predicates are variables representing hidden or unknown aspects that influence observed phenomena but are not directly observable. Identifying these latent predicates is crucial for comprehensive understanding and accurate decision-making. In our approach, the model learns rules with the template as Eq. (1) that include latent predicates, treating them as placeholders for unknown relationships or factors. As the model identifies and integrates these latent predicates into the rules, it uncovers hidden structures within the data.

Once the rules are established, we perform **post-hoc labeling** of the latent predicates by interpreting their roles in the rule structure. This process allows us to assign meaningful labels to previously unknown predicates based on their discovered relationships and functions.

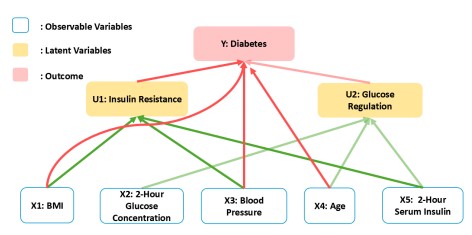 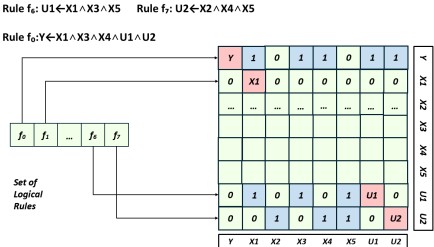

(a) Example of Latent Variables in Rule Discovery    (b) Visualization of Rule Embedding

Figure 1: Rule Embedding

# 4 MODEL:RECURRENT NEURO-SYMBOLIC FORWARD CHAINING NETWORK

Consider a classification problem where the goal is to map input binary feature $X$ to an output label $Y$, with the label set $Y$ consisting of $L$ distinct labels. The model's objective is to learn a set of logical rules that explain how each label can be inferred based on evidence from $X$. Additionally, our model accounts for latent variables–unobserved predicates that influence the classification decision. These latent variables are inferred through the rule-learning process, allowing the model to uncover hidden relationships in the data. This approach enables the induction of interpretable rules, linking both observed and latent factors to the predicted label, enhancing the model's predictive power and interpretability.

## 4.1 MODEL PREPARATION: PRETRAINED PREDICATE EMBEDDINGS

As a preparation for our RNS-FCN model, we will introduce predicate embeddings for $X$, $Y$, and latent variables. Note that these predicate embeddings will be frozen during the training.

**Predicates Embedding for $X$**    Denote $X = \{X_1, \ldots, X_n\}$ as a set of predicate variables. Each predicate $X_i \in \{1, 0\}$ is a Boolean random variable with a prespecified meaning. We associate each of the True and False states of the predicate with two paired predicate embeddings, denoted as $K_{X_i}$ and $K_{\neg X_i}$, with $K_{X_i}, K_{\neg X_i} \in \mathbb{R}^d$ and $K_{X_i} = -K_{\neg X_i}, \forall X_i$. In this way, for each states of the predicate, we can find their embedding from $K_X = [K_{X_1}, \ldots, K_{X_n}]$ and $K_{\neg X} = [K_{\neg X_1}, \ldots, K_{\neg X_n}]$. These embeddings can be pretrained to capture the meaning and relationships between predicates. They act like a lookup dictionary, providing a fixed representation for each state of the predicate.

**Predicate Embeddings for Latent Variables**    Our model is designed to discover new predicates, a crucial capability in scenarios where not all predicates are known a priori. Let $U = \{U_1, \ldots, U_m\}$ represent a set of latent or yet-to-be-invented predicates with initially *undefined* meanings. The number of these unlabeled predicates, $m$, is a hyperparameter that can be adjusted based on the specific application or data complexity. Each latent predicate $U_i \in \{1, 0\}$, and its True and False states are also linked to a paired embedding vectors, denoted as $K_{U_i}, K_{\neg U_i} \in \mathbb{R}^d$ with $K_{U_i} = -K_{\neg U_i}, \forall U_i$. These embeddings act as placeholders for these unlabeled predicates. We aim to interpret these latent predicates in a posthoc way as our model uncovers logical rules.

**Predicate Embeddings for $Y$**    In the classification problem, each label $y \in Y$ can be represented as an embedding vector (for example, a one-hot vector).

We freeze the predicate embeddings during rule learning, and each rule is represented as a rule embedding. All the rule embeddings are treated as model parameters and are constructed as conjunctive combinations of predicate embeddings. Our model learn to select the best matching predicates to fill the rule's structure. The rule embeddings are optimized via gradient descent in a differentiable manner, allowing efficient learning. This separation—fixed predicate embeddings and learnable rule embeddings—enables the model to focus on discovering relevant logical rules

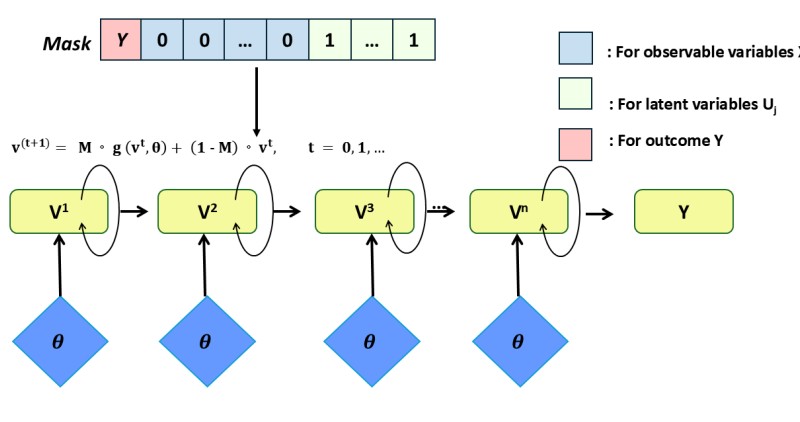

Figure 2: Recurrent Unit: Backbone for Logical Reasoning

## 4.2 MODEL BACKBONE: RECURRENT UNIT

Our proposed RNS-FCN models rule embeddings as trainable parameters within a **recurrent neural network** architecture. Unlike traditional RNNs that process sequences of data, each **recurrent unit** in RNS-FCN is responsible for performing logical reasoning over predicates (symbolic variables). At each step, the recurrent unit applies the learned rules to update the model's understanding of the logical relationships in the data. This process continues iteratively, refining inferences until a stable state is reached. All rule embeddings are optimized end-to-end through backpropagation across the recurrent units, enabling the model to learn complex logical patterns.

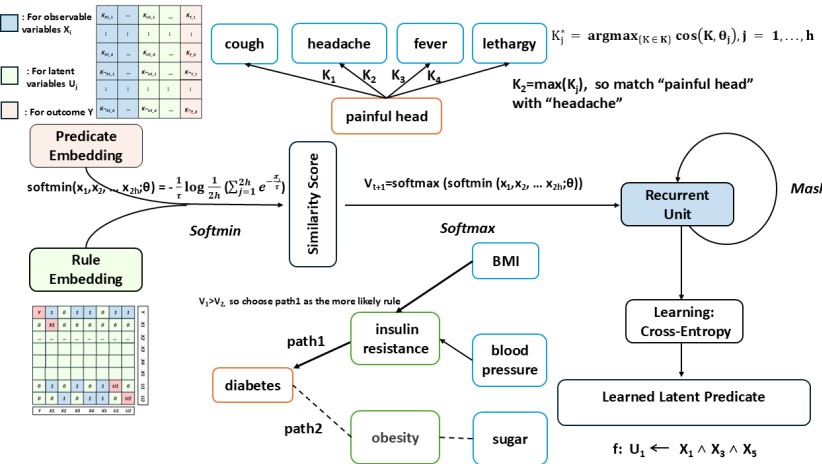

Figure 3: Architecture of RNS-FCN with Examples

**Architecture of RNS-FCN** The architecture of RNS-FCN is similar to that of RNNs in that it uses **recurrent units**, but the goal of each unit is to perform logical reasoning based on the current state of the inferred facts given the learned rules.

- **Hidden state**: Let $v^t \in [-1, 1]^{2(n+m+L)}$ represent the hidden state of the recurrent unit indexed by $t$, where each element of $v^t$ is

$$[X_1, \neg X_1, \dots, U_1, \neg U_1, \dots, Y_1, \neg Y_1, \dots] = [x_1, 1 - x_1 \dots, u_1, 1 - u_1 \dots, y_1, 1 - y_1, \dots]$$

  with $x_1, \dots, x_n \in \{0, 1\}$ denoting observable and grounded predicate facts, $u_1, \dots, u_m$ representing the current inferred states of the latent predicate, and $y_1, \dots, y_L$ indicating the current

inferred states of the output label. For observed predicates, its value is $1$ or $0$. For unobserved predicates and labels, we initialize their corresponding elements with a random guess, typically set to $\frac{1}{2}$. Importantly, all unlabeled predicates and the predictive labels can only be inferred through the rule-learning process.

- **Mask**: Let $\boldsymbol{M} \in \{0,1\}^{2(n+m+L)}$ be a binary mask that indicates which elements of $\boldsymbol{v}^t$ need to be inferred. Specifically, for the observed predicates, the corresponding elements of $\boldsymbol{M}$ are set to 0, meaning these values will remain fixed during the inference process. Alternatively, the corresponding elements of $\boldsymbol{M}$ are set to 1, indicating that these elements of $\boldsymbol{v}^t$ are unobserved and must be inferred through the rule-learning process. In this way, the mask $\boldsymbol{M}$ serves as a guide, ensuring that only the unknown facts are updated during the inference phase.

- **Recurrent unit**: At each step, the RNS-FCN applies the current logical rules to update its belief about the state of predicates. The recurrent units operate similarly to RNNs, but instead of processing raw data, they process logical inferences. The inference process is facilitated by employing an RNN-type model

$$\boldsymbol{v}^{t+1} = \boldsymbol{M} \circ g(\boldsymbol{v}^t, \Theta) + (1 - \boldsymbol{M}) \circ \boldsymbol{v}^t \qquad t = 0, 1, \ldots \tag{2}$$

where, $g\left(\boldsymbol{v}^t, \Theta\right)$ is a mapping from $\mathbb{R}^{n+m} \rightarrow \mathbb{R}^{n+m}$, where $\Theta$ are the model parameters representing the learned logical rules, and $\boldsymbol{v}^t$ is the state of the predicate vector at time step $t$. Note that each recurrent layer indexed by $t$ shares the same set of model parameters $\Theta$. The operator $\circ$ denotes element-wise multiplication. We will recursively perform the updates until $\boldsymbol{v}^t$ remains unchanged or converges to a fixed point.

**Specification of $\Theta$ in Eq. (2)**  The parameter $\Theta = [\Theta_f]_{f \in \mathcal{F}}$ represents the set of logical rules we aim to learn, denoted as $\mathcal{F} := \{f\}$. Each matrix $\Theta_f$ encodes a logical rule that can either predict potential labels in $Y$ (in a classification task) or infer latent predicates. Based on the general rule structure shown in Eq. (1), each rule is parameterized by a matrix $\Theta_f \in \mathbb{R}^{d \times (h+1)}$, where $d$ is the dimensionality of the predicate embeddings, and $h$ is the number of predicates in the body of the rule. The first column of $\Theta_f$ corresponds to the embedding of the head predicate $Q$, while the remaining $h$ columns represent the embeddings of the body predicates $P_1, P_2, \ldots, P_h$.

In our model, for classification tasks, each label $Y$ has at least one corresponding rule $\Theta_i$ where the first column (head predicate) is fixed to the embedding of that label. The goal is to learn the embeddings of the body predicates that best explain how the label $Y$ can be inferred from the input data $\boldsymbol{X}$.

Similarly, for latent predicates, each latent predicate is associated with at least one rule $\Theta_i$, where the head predicate column is fixed to the embedding of the latent predicate. The task here is to learn the body predicate embeddings that allow the model to deduce the latent predicate from the observed data.

In summary, the **head predicate embeddings** in $\Theta$ are fixed, representing either labels or latent predicates. The learning task focuses on discovering the **body predicate embeddings** to complete the logical rules, enabling the model to infer both labels and latent predicates from the observed data. Increasing the size of $\Theta$ either by adding more rules or extending the length of each rule affects the model's complexity and expressiveness. The number of rules and the length of each rule are hyperparameters that determine the model's capacity to capture intricate patterns in the data.

**Specification of $g(\cdot, \Theta)$ in Eq. (2)**  The mapping function $g(\cdot, \Theta)$ in Eq. (2) utilizes logical rules to iteratively update beliefs about the state of latent predicates and labels until convergence is achieved.

For a general rule embedding $\Theta_f = [\theta_0, \ldots, \theta_h] \in \mathbb{R}^{d \times (h+1)}$, where the first column $\theta_0 \in \mathbb{R}^d$ represents the fixed embedding for the head predicate, the rule is used to infer a latent predicate or label $v$. Each column $\theta_j (j = 1, \ldots, h)$ of the body predicates in the rule embedding is matched with a corresponding predicate embedding. This matching is achieved by finding the predicate embedding most similar to $\theta_j$ using cosine similarity:

$$K_j^* = \operatorname*{argmax}_{K \in \boldsymbol{K}} \cos(K, \theta_j), \quad j = 1, \ldots, h \tag{3}$$

where $\boldsymbol{K} = \boldsymbol{K}_X \cup \boldsymbol{K}_{\neg X} \cup \boldsymbol{K}_U \cup \boldsymbol{K}_{\neg U}$ represents the set of all available predicate embeddings. The inverse mapping $I(K)$ maps a predicate embedding $K \in \mathbb{R}^d$ back to its corresponding index. Specifically:

- For an index $i \in \{1, \ldots, 2n\}$, the index maps to the positive observed predicate embedding $\boldsymbol{K}_X$ and its corresponding negative observed predicate embedding $\boldsymbol{K}_{\neg X}$.

- For an index $i \in \{2n+1, \ldots, 2n+2m\}$, the index maps to the latent predicate embedding $\boldsymbol{K}_U$ and its corresponding negated latent predicate embedding $\boldsymbol{K}_{\neg U}$.

Thus, indices $1, \ldots, 2(n+m)$ correspond to $2n+2m$ predicate embeddings, capturing both positive and negative signs.

Once the best matching predicates $K_j^*$ for each $\theta_j (j = 1, \ldots, h)$ are determined, we update the inferred states for the latent predicate or label $v_{\text{head}}^{t+1} \in \boldsymbol{v}^{t+1}$ as follows:

$$v_{t+1} = \prod_{j=1,\ldots,h} \cos\left(K_j^*, \theta_j\right) \prod_{j=1,\ldots,h} \boldsymbol{v}^t \left(I\left(K_j^*\right)\right)$$

To address the potential issue of diminishing values over iterations, we can use the min function instead:

$$v_{t+1} = \min_{j=1,\ldots,h} \left\{\cos\left(K_j^*, \theta_j\right), \boldsymbol{v}^t \left(I\left(K_j^*\right)\right)\right\}$$

However, to make this function differentiable, we approximate the min function using the softmin function. For each $j$, there are two terms: $\cos\left(K_j^*, \theta_j\right)$ and $\boldsymbol{v}^t \left(I\left(K_j^*\right)\right)$. The softmin function can be formulated as:

$$\text{softmin}\left(x_1, \ldots, x_{2h}; \Theta\right) = -\frac{1}{\tau} \log \frac{1}{2h} \left(\sum_{j=1}^{2h} e^{-x_j/\tau}\right)$$

where $x_j$ represents either $\cos\left(K_j^*, \theta_j\right)$ or $\boldsymbol{v}^t \left(I\left(K_j^*\right)\right)$, and $\tau$ is a temperature parameter controlling the smoothness of the approximation. As $\tau$ approaches 0, the softmin function approximates the behavior of the hard min function.

If multiple rules can be used to infer $v_{t+1}$, we apply the softmax function over the results of each rule. This ensures that the most likely rule's output is favored:

$$v_{t+1} = \text{softmax}\left(\text{softmin}\left(x_1, \ldots, x_{2h}; \Theta\right) \text{ for all rules }\right)$$

Here, $\text{softmax}\left(x_1, \ldots, x_R\right)$ ensures that if any rule provides strong evidence for the latent predicate, the inferred value $v_{t+1}$ will be high.

### 4.3 MODEL LEARNING: BACKPROPAGATION

In the RNS-FCN model, backpropagation is used to train the network by computing the gradient of the loss function with respect to the rule embeddings $\Theta$. This allows the model to adjust its embeddings and improve prediction accuracy.

The loss function quantifies the discrepancy between the predicted latent label probabilities and the actual labels. For multiclass classification, we use the Cross-Entropy Loss. Given $\boldsymbol{v}^T$ as the final output vector of the network (representing the predicted class probabilities for latent predicates) after $T$ recurrent unit iterations and $\boldsymbol{y}$ as the true class labels, the loss function is formulated as:

$$J = -\sum_{i=1}^{N} \sum_{l=1}^{L} y_{i,l} \log\left(\hat{y}_{i,l}\right)$$

where $y_{i,l}$ is a binary indicator (0 or 1) if class label $l$ is the correct classification for sample $i$, $\hat{y}_{i,l}$ is the predicted probability of class $l$ for sample $i$, and $L$ is the number of classes. This loss function measures how well the predicted probabilities $\boldsymbol{v}_y^T$ align with the actual labels.

During backpropagation, the gradient of the loss function with respect to the rule embeddings $\Theta$ is computed using the chain rule:

$$\frac{\partial J}{\partial \Theta} = \frac{\partial J}{\partial \boldsymbol{v}_y^T} \cdot \frac{\partial \boldsymbol{v}_y^T}{\partial \Theta}$$

Here, $\boldsymbol{v}_y^T$ represents the predicted latent label probabilities after $T$ iterations of the recurrent units. The rule embeddings $\Theta$ are updated in the direction opposite to the gradient, scaled by a learning rate $\alpha$ :

$$\Theta := \Theta - \alpha \frac{\partial J}{\partial \Theta}$$

In summary, in the RNS-FCN model, the Cross-Entropy Loss is used to train the network to infer latent predicates or labels from the rule-based embeddings. The forward computation involves selecting the most matching predicate embeddings based on cosine similarity, filling in the slots of the rule embedding, and then updating the inferred states. After $T$ recurrent unit iterations, the hidden states converge to a stable representation, which is used for the final prediction. The loss function is applied to the final predicted outputs, which are derived from the latent predicates. By using backpropagation, we optimize the rule embeddings $\Theta$ to improve the accuracy of the latent label predictions.

## 5 Experiments

### 5.1 Synthetic Data Experiments

We conducted simulation experiments using synthetic data with known ground truth rules to evaluate the ability of our proposed method *RNS-FCN* in learning rules with latent variables. In this setting, we generate a set of synthetic data using the following ground truth rules:

$$X_3 \leftarrow X_1 \wedge X_2, X_4 \leftarrow X_0 \wedge X_1, Y \leftarrow X_3 \wedge X_5, Y \leftarrow X_2 \wedge X_4. \tag{4}$$

In this case, $X_1$, $X_2$ and $X_5$ are observed variables. $X_3$ and $X_4$ are unobserved latent variables. Further, Y is the observed label. All the variables and label Y are binary categorical variables. It can use 1 to indicate affirmation and 0 to indicate negation. When we generate the synthetic dataset by the following probability:

$$P(\{X_0 = 1\} = 0.6), P(\{X_1 = 1\} = 0.8), P(\{X_2 = 1\} = 0.7), P(\{X_5 = 1\} = 0.7), \tag{5}$$

the rules learned by *RNS-FCN* are:

$$X_4 \leftarrow X_0 \wedge X_1 \wedge X_2, X_3 \leftarrow X_5 \wedge X_2, Y \leftarrow X_3, Y \leftarrow X_2 \wedge X_4. \tag{6}$$

Moreover, when we use 80% of the dataset as the training set and 20% as the test set, the predicting accuracy in the test set is 0.77, and the F1 score in the test set is 0.81. This synthetic data experiment demonstrates our proposed framework's ability to recover ground truth rules from data with latent variables and make accurate predictions at the same time.

### 5.2 Real-World Data Experiments

#### 5.2.1 Datasets Discreption

To demonstrate the practical value of our method, we validated its effectiveness in learning complex rules on two representative datasets.

- **Heloc** is a dataset collected from FICO. The fundamental task is to use the information about the applicant in their credit report to predict whether they will repay their HELOC account within two years. For each applicant, 154 factors are contained in the dataset.

- **Adult**, also known as the "Census Income" dataset, is a famous dataset in social science. The corresponding task for this dataset is predicting whether an individual's annual income exceeds $50,000 dollars a year based on census data.

#### 5.2.2 Data Preprocessing

To evaluate our proposed method *RNS-FCN*'s ability to learn under invisible content, we removed certain variables to simulate a scenario with latent variables. Specifically, for dataset **Heloc**, we delete 70 of all the 154 factors. The rule-based prediction model is fitted by the rest of the variables in our experiment settings. The prediction accuracy in testing sets demonstrates model performance. For dataset **Adult**, we delete 100 of all the 128 factors. The rule-based prediction model is fitted

by the rest of the variables. Due to the severe class imbalance in the labels, with more than three-quarters of the samples labeled as "Bad," we performed sampling on the dataset to construct a new dataset with a 1:1 ratio of positive to negative samples. Moreover, in each setting, 80% of the data is split into a training set, and the rest of 20 % is split into the testing set. The prediction accuracy in testing sets demonstrates model performance.

### 5.2.3 BASELINE MODELS

- **RIPPER** (Cohen, 1995) is a rapid rule induction algorithm that constructs rules in an iterative way, employing a separate-and-conquer approach.

- **BRCG** (Dash et al., 2018) is an integer program, which is formulated to trade classification accuracy for rule simplicity optimally. Column generation is used to efficiently search over an exponential number of candidate clauses (conjunctions or disjunctions) without needing heuristic rule mining.

- **LEN** (Barbiero et al., 2022) is a novel end-to-end differentiable approach enabling the extraction of logic explanations from neural networks using the formalism of First-Order Logic. The method relies on an entropy-based criterion that automatically identifies the most relevant concepts.

- **DR-NET** (Qiao et al., 2021) is a new paradigm for learning a set of independent logical rules in disjunctive standard form as an interpretable model for classification. The problem of learning an interpretable decision rule set is considered training a two-layer neural network.

- **RRL** (Wang et al., 2021) can automatically learn interpretable non-fuzzy rules for data representation and classification. The non-differentiable RRL is projected to a continuous space and optimized by a novel training method, called Gradient Grafting, that can directly optimize the discrete model using gradient descent.

- **Black-Box Model** We use a neural network-based predictor as a black-box predictor. We adopt a two-layer Multilayer perceptron (MLP) as the predictor in this setting (Taud & Mas, 2018). Generally, neural networks based methods have excellent predictive performance but are not as explainable as rule-based methods (Madsen et al., 2022).

| Method | Heloc | | Adult | |
|---|---|---|---|---|
| | Accuracy | F1 | Precision | F1 |
| RNS-FCN | **0.68** | **0.67** | 0.54 | 0.52 |
| RIPPER | 0.52 | 0.00 | 0.50 | 0.00 |
| BRCG | 0.66 | 0.60 | 0.72 | **0.76** |
| LEN | 0.54 | 0.64 | 0.50 | 0.75 |
| DR-NET | 0.67 | 0.59 | 0.70 | 0.76 |
| RRL | 0.66 | 0.66 | **0.77** | 0.52 |
| Black-Box | 0.71 | 0.68 | 0.76 | 0.75 |

Table 1: Performance of seven methods on two datasets

### 5.2.4 NUMERICAL PERFORMANCE ANALYSIS

We use accuracy and F1 score of binary classification on the test set as the model's evaluation metrics. Experiment results in the deleted **Heloc** dataset demonstrate that our proposed method RNS-FCN outperforms all the rule-based baselines on both preciting accuracy and F1 score. The experimental results indicate that, among rule-based methods, our proposed approach achieves the best predictive performance on datasets with missing variables. Further, our method's performance is close to that of the black-box model, indirectly confirming its effectiveness and practicality. This also highlights the trade-off between interpretability and performance.

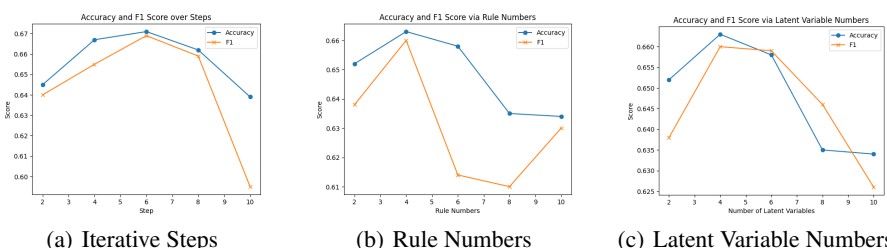

(a) Iterative Steps  (b) Rule Numbers  (c) Latent Variable Numbers

Figure 4: Hyper-Parameters Analysis

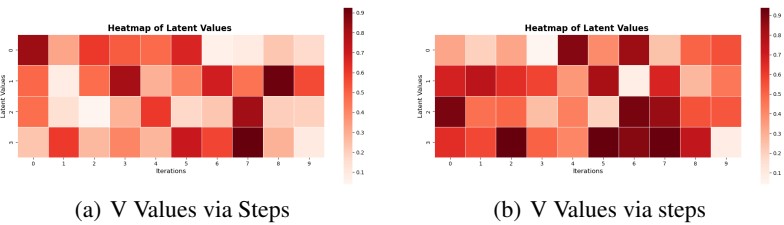

(a) V Values via Steps  (b) V Values via steps

Figure 5: Evaluation for Iteration Steps

### 5.2.5 SENSITIVE ANALYSIS FOR HYPER-PARAMETERS

The most critical hyper-parameters in our proposed framework are the number of iterations, the number of rules for inference y and the number of latent variables. We conducted experiments on the dataset **Heloc** to test our model's sensitivity to hyperparameters and select the most suitable ones. The model's performance under different hyperparameter settings was also evaluated using prediction accuracy and F1 score as the evaluation metrics. The result in 4(a) shows that our method achieve its best performance when Iterate six times. The result in 4(b) shows that our method performs best when there are four rules. Moreover, the result in 4(c) demonstrates that our model achieves the peak performance when setting four latent variables. This is because when there are too few latent variables, the model fails to capture enough of the missing information, leading to underfitting. Conversely, when there are too many latent variables, they introduce noise that can disrupt the prediction, resulting in overfitting. Furthermore, Consequently, and the default literations is six times, the default number of rules to infer y is four, and the default number of latent variables is also four.

### 5.2.6 VISUALIZATION FOR ITERATIONS

In this setting, we set the number of iterations to 10 and the number of latent variables to 4 5. We examined how the values corresponding to the latent variables change with the number of iterations. We visualize the changes in the v-values of two sample points across the iterations. The experimental results show that the v values of different samples tend to converge around the 6th or 7th iteration, but as the iterations approach 10, they start to diverge. This indicates that the v-values are in an optimal state around the 6th-7th iteration, and further iterations disrupt this balance. This aligns with our experimental findings, where the model performs best after 6 iterations.

## 6 CONCLUSIONS

We addresses a fundamental limitation of traditional rule induction methods: the inability to infer and discover latent predicates that are not directly observable by bridging the gap between explicit rule induction and latent variable modeling, offering a powerful tool for uncovering hidden structures and generating new concepts.

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
