# OpenReview forum: "Inferring the Invisible: Recurrent Neuro-Symbolic Forward Chaining Network"
_ICLR.cc/2025/Conference — ICLR 2025 Conference Withdrawn Submission_

### Official Review · Reviewer_y7NX · 2024-10-28

**Soundness:** 3
**Presentation:** 2
**Contribution:** 3
**Rating:** 3
**Confidence:** 4

**Summary:**

The authors consider the problem of discovering predictive symbolic rules for binary (predicate) data. They present a novel and creative solution that shows how to adapt the principles of learning recurrent neural networks to this problem using a specific recurrent unit that can be interpreted as symbolic rules. In their architecture, they specifically propose to model latent (symbolic) variables to account for unobserved, complex factors that govern a prediction. Proposing a gradient-based solution for learning these units, they further show how their approach yields reasonable symbolic rules in synthetic and real world data.

**Strengths:**

- The proposed architecture seems novel and thought-through.
- The paper is well written and easy to follow.
- The solution seems creative and promising.

**Weaknesses:**

- The motivation is not completely clear; the critical latent condition mentioned in the introduction are neither named nor are such latent conditions presented in the (real world) data experiments. The paper would greatly benefit from explicit examples (e.g. for the mentioned healthcare scenario) of such latent factors that can not be directly obtained as data (are there for example factors that are impossible to measure in a patient, or too costly to do so?). While the presented experiments do make sense, as holding out factors allows for quantitative evaluation, applying the method on real world data to infer *new* (i.e., nothing that actually has been measured) latent factors would make for a strong application.
- The results are unfortunately a bit underwhelming: despite being very small and without noise, the correct set of rules could not be recovered in the synthetic data study. The performance on real data (see Adult data set in Table 1), is comparably quite bad to existing work. The meaning of inferred latent values (e.g., are they related to the removed factors from the data) is not analysed/evaluated.
As concrete steps of action for synthetic data, I would suggest to extend the experiments to study more datasets, varying the number of rules, adding noise to experiments and compare to existing work. This would on the one hand give a better understanding of the capabilities of the presented approach, but on the other hand also give better feeling of how the method behaves compared to existing work: My gut feeling is that existing work will perform much worse than RNS-FCN on the synthetic data, where there are complex hidden factors. It would hence put your method in a different (positive) light. Regarding the real data, I would propose to investigate whether latent factors are actually used in Adult,  from experience, this dataset is actually quite simple, so modeling latents might not be of use here. A critical discussion of these results would help.

**Questions:**

- see Weaknesses, and
- How does the post-hoc labeling of latents work? I did not see it being explained for synthetic data experiments nor actually used in the real world data scenario.
- How close are the predicted rule embedding to the predicate embeddings? That is, when you assign a predicate to a rule body, how far away (e.g. measured as cosine similarity) is it? Do you make a large error in the assignment when you look at some actual data?
- What is the runtime in terms of complexity, what is the average number of recurrences $T$ until convergence, and what is the effective runtime in the considered experiments?
- Are there cases when $v^t$ does not converge? Why or why not?
- What is the difference in Figure 5a and b? Is it the two different data sets?
- Why was the sampling of data to get 1:1 negative/positive label ratio necessary? Is your method not robust to this? Please discuss.
- How does your method compare to Diffnaps [1], one of the most recent predictive rule-mining methods?


Further details:
- Please fix the citations!
- In the intro, you mention that for traditional deep learning, increased depth refers to larger networks but without a logical reasoning process guiding the layers. Esp. in computer vision, this is actually not true. There is plenty of evidence from the XAI community that layers build more and more complex concepts with depth, combining earlier features. You further say that your model “[pushes] beyond surface-level pattern recognition engaging in multi-step inferencing akin to human thought,[…]”, which is very far fetched and not evident in your real world data study. I would strongly suggest you tone down the intro a bit.
- The paper would benefit from a proper discussion.

[1] NP Walter et al. Finding Interpretable Class-Specific Patterns through Efficient Neural Search AAAI 24

---

### Official Review · Reviewer_1tqb · 2024-11-01

**Soundness:** 1
**Presentation:** 2
**Contribution:** 2
**Rating:** 3
**Confidence:** 3

**Summary:**

It seems this work aims to discover latent predicates from task and construct rules or concepts between latent predicates. But it is written in a less professional way, it is vague what some of the terminologies are referring to. For example "underlying factors", "it represents deeper reasoning". So it is kind of challenging to understand what this work is actually doing.

**Strengths:**

1. The framework is interesting, although similar to a previous method.

**Weaknesses:**

1. Code is not provided. The only file in the supplementary pack is ``untitled.ipython'', which doesn't look like valid code. The authors generated their own synthetic dataset. The dataset is not provided either.
2. Citation style is incorrect.
3. The proposed framework is very similar to [1]. You both have a forward inference stage and backward rule refining stage. You both defined invented predicate. Fig1b is similar to their framework too. Please consider distinguishing the two in your paper, and compare with it.
4. Too few experiments. Very old baselines. The proposed model is not the best, but marked as the best in Table 1. The method doesn't seem to work effectively according to the results.

[1] Lu, et al. "R5: Rule Discovery with Reinforced and Recurrent Relational Reasoning." ICLR 2022.

**Questions:**

See Weaknesses.
1. Is the term "Latent predicate" firstly invented in this paper? If no, can you cite the original papers using this term? If yes, can you explain in more details why it is necessary?

**Details Of Ethics Concerns:**

This paper looks to be LLM written. Code is not provided. None of the figures are referred in the main text. Further review may needed.

---

### Official Review · Reviewer_Ct9Q · 2024-11-04

**Soundness:** 3
**Presentation:** 2
**Contribution:** 2
**Rating:** 5
**Confidence:** 3

**Summary:**

In this paper, the authors propose RNS-FCN, a Recurrent Neuro-Symbolic Forward Chaining Network designed for logical reasoning. This approach effectively addresses challenges posed by latent, unobservable variables. With its recurrent architecture, RNS-FCN is capable of simulating a human-like thought process, enabling iterative reasoning and refinement as new evidence emerges. The authors evaluate the model’s effectiveness through experiments on both simulated and real-world datasets, accompanied by analyses that provide insights into the model's behavior and underlying mechanisms.

**Strengths:**

1. The proposed RNS-FCN introduces a novel model architecture, featuring well-defined variables and well-justified design choices, including a recurrent unit and backpropagation for model learning. The incorporation of the softmin function is smart choice, as it prevents diminishing values over iterations while preserving differentiability.

2. The authors conducted both simulations on synthetic data and evaluations on real-world datasets to validate the effectiveness of the proposed architecture. The simulation experiments are helpful for readers to gain a straightforward and intuitive understanding of the proposed approach.

3. The proposed method’s ability to handle unobservable latent variables adds practical value, as it addresses hidden factors often present in real-world reasoning tasks.

**Weaknesses:**

1. The performance improvement on the two real-world datasets, Heloc and Adult, appears to be incremental. Notably, the accuracy and F1 score of RNS-FCN on the Adult dataset are inferior to several of the compared baselines, raising concerns about the effectiveness of the proposed approach. Unfortunately, the authors do not provide any explanations regarding the performance on this dataset. A thorough analysis in this area would be beneficial for understanding the conditions under which the proposed approach may fail and for identifying potential improvements.

2. It remains unclear how this approach is more interpretable than neural network baselines. The introduction of nonlinear functions, such as softmin, and the optimization through backpropagation may complicate interpretability rather than enhance it.

3. Several experimental setup details, including hyperparameter tuning and selection, are lacking in the paper. Is there a validation set used for this purpose? Additionally, how was the embedding dimensionality \( d \) determined?

**Questions:**

1. In line 197, the authors mention that the predicate embeddings can be pretrained to capture the meanings and relationships between predicates. How can this pretraining be done? Any details on the pretaining stage?

2. How do you define the conjunctive combinations of predicate embeddings?

3. Is it optimal to initialize unobserved predicates with a random guess? Is there any analysis supporting this choice?

4. Why is the number of iterations considered a critical hyperparameter? Is there a validation set used to assess its impact?

5. What is the computational complexity of the proposed architecture?

---

### Official Review · Reviewer_cNuJ · 2024-11-08

**Soundness:** 2
**Presentation:** 2
**Contribution:** 2
**Rating:** 3
**Confidence:** 4

**Summary:**

The authors addresses a fundamental limitation of traditional rule induction methods: the inability to infer and discover latent predicates that are not directly observable by bridging the gap between explicit rule induction and latent variable modeling, offering a powerful tool for uncovering hidden structures and generating new concepts.

**Strengths:**

1. The proposed framework bridges the gap between traditional rule induction and the modeling of latent variables

2. The RNS-FCN framework provides a robust tool for discovering hidden structures and generating new concepts, offering a comprehensive solution for tasks that require deep understanding and adaptability.

3. This end-to-end approach enhances our ability to uncover latent factors, making it invaluable for applications ranging from user behavior analysis to advanced decision-making systems in AI.

**Weaknesses:**

1.	Limited Technical Contribution: The technical novelty of the paper is limited. For instance, Section 4.3 merely covers standard backpropagation, which does not warrant inclusion as a significant contribution in the main paper.

2.	Insufficient Evaluation Datasets: The method’s effectiveness is demonstrated on only two small datasets, which is inadequate for validating its general applicability and robustness.

3.	Lack of Ablation Study: An ablation study is necessary to verify the contribution and impact of each component within the proposed framework.

**Questions:**

See Weaknesses

---

### Note · Authors · 2024-11-27

**Comment:**

Thank you for your thoughtful reviews. We have learned a lot from your feedback and will try to improve our work based on your insights in the future.

**Withdrawal Confirmation:**

I have read and agree with the venue's withdrawal policy on behalf of myself and my co-authors.